# Well-Being, Physical Activity, and Social Support in Octogenarians with Heart Failure during COVID-19 Confinement: A Mixed-Methods Study

**DOI:** 10.3390/ijerph192215316

**Published:** 2022-11-19

**Authors:** Elena Marques-Sule, Elena Muñoz-Gómez, Luis Almenar-Bonet, Noemi Moreno-Segura, María-Cruz Sánchez-Gómez, Pallav Deka, Raquel López-Vilella, Leonie Klompstra, Juan Luis Cabanillas-García

**Affiliations:** 1Physiotherapy in Motion, Multispeciality Research Group (PTinMOTION), 46010 Valencia, Spain; 2Department of Physiotherapy, University of Valencia, 46010 Valencia, Spain; 3Heart Failure and Transplantation Unit, Department of Cardiology, Hospital Universitario y Politécnico La Fe, 46026 Valencia, Spain; 4Centro de Investigación Biomédica en Red de Enfermedades Cardiovasculares (CIBERCV), Instituto de Salud Carlos III, 28029 Madrid, Spain; 5Department of Medicine, University of Valencia, 46010 Valencia, Spain; 6Department of Didactics, Organization and Research Methods, University of Salamanca, Paseo de Canalejas 169, 37008 Salamanca, Spain; 7College of Nursing, Michigan State University, East Lansing, MI 3078, USA; 8Department of Health, Medicine and Caring Sciences, Linkoping University, 4566 Linkoping, Sweden

**Keywords:** heart failure, COVID-19, confinement, well-being, physical activity, mixed-methods study

## Abstract

Background: This study aimed to compare well-being and physical activity (PA) before and during COVID-19 confinement in older adults with heart failure (HF), to compare well-being and PA during COVID-19 confinement in octogenarians and non-octogenarians, and to explore well-being, social support, attention to symptoms, and assistance needs during confinement in this population. Methods: A mixed-methods design was performed. Well-being (Cantril Ladder of Life) and PA (International Physical Activity Questionnaire) were assessed. Semi-structured interviews were performed to assess the rest of the variables. Results: 120 participants were evaluated (74.16 ± 12.90 years; octogenarians = 44.16%, non-octogenarians = 55.83%). Both groups showed lower well-being and performed less PA during confinement than before (*p* < 0.001). Octogenarians reported lower well-being (*p* = 0.02), higher sedentary time (*p* = 0.03), and lower levels of moderate PA (*p* = 0.04) during confinement. Most individuals in the sample considered their well-being to have decreased during confinement, 30% reported decreased social support, 50% increased their attention to symptoms, and 60% were not satisfied with the assistance received. Octogenarians were more severely impacted during confinement than non-octogenarians in terms of well-being, attention to symptoms, and assistance needs. Conclusions: Well-being and PA decreased during confinement, although octogenarians were more affected than non-octogenarians. Remote monitoring strategies are needed in elders with HF to control health outcomes in critical periods, especially in octogenarians.

## 1. Introduction

Heart failure (HF) is a major clinical and public health problem [1]. HF constitutes a complex debilitating syndrome with significant health consequences that trigger high burdens of mortality and hospitalization, particularly among those aged 65 and older [2].

The COVID-19 pandemic has caused an alteration and restructuration of healthcare systems. The routine of care of HF patients has abruptly changed compared to pre-COVID-19 in terms of an increase in in-person outpatient visits and an increase in telehealth-based programs [3]. Moreover, because government-approved nationwide confinement may have led to a decrease in physical activity (PA), which could also lower well-being, there may be serious consequences for cardiometabolic morbimortality in HF patients. Therefore, COVID-19 represents a serious threat for HF patients, even more in elders with HF [4,5,6]. On the other hand, both confinement and fear of contagion may have reduced social support and assistance for elder patients with HF during COVID-19 confinement [7].

In order to mitigate these consequences, promotion of self-management and optimization of clinical assistance of patients with HF, as well as the use of measures such as telehealth, telerehabilitation, mobile applications, and electronic heart rate devices, have been employed [8,9]. Previous trials have found several limitations such as the lack of physical examination, depersonalization of healthcare, and lack of familiarity with the platform [10]. Moreover, there are groups of population that seem not to be ready to use these telematic measures, due to very old age, poor hearing, cognitive dysfunction, language barriers, or limited education, which may require the assistance of a family member or a caregiver [5].

Previous quantitative studies have analyzed how confinement affected patients with HF, in terms of fear to visit the hospital [3], differences in physical activity [11,12], mental health [13], safety of telehealth programs [14], etc. Few qualitative studies have been performed in order to characterize the value of technology in supporting caregiving for individuals with HF [15], examine the caregiving experiences and coping strategies of older adults with HF during the ongoing pandemic [16], and explore patients’ and clinicians’ experiences of managing HF during COVID-19 pandemic [17]. However, to our knowledge, no mixed-methods studies have been conducted in this regard. Mix-methods designs offer the possibility for participants to give their opinion about issues that may not have been considered in pre-set quantitative questionnaires. In addition, the literature regarding delivery of health care has highlighted the role of patient-reported outcomes and also of experience measures [10].

Even though previous studies have been performed in patients with HF, most of them take into account the middle-aged population or, when assessing the elderly, ages range from 60 to 80 years, thus information regarding adults older than 80 years old is scarce [11,13,14,16,18,19]. Therefore, further research is needed to focus on the clinical conditions of the octogenarian population and to analyze the differences between octogenarians and non-octogenarians.

This study aimed to (1) compare well-being and PA before and during COVID-19 confinement in older patients with HF; (2) compare well-being and PA during COVID-19 confinement in octogenarians and non-octogenarians with HF; and (3) explore well-being, social support, attention to HF symptoms, assistance needs, and suggestions to improve care during COVID-19 confinement in this population.

## 2. Materials and Methods

### 2.1. Participants and Setting

The sample consisted of 120 participants. Participants were recruited at an outpatient clinic and were assessed between November 2019 and April 2020. Inclusion criteria included: (1) diagnosis of HF, (2) age ≥ 60 years, and (3) being cognitively capable of completing the assessments. Written informed consent was obtained before participation. The principles of voluntariness and confidentiality were respected. All participants were informed about the objectives and procedures of the study. All procedures were conducted in accordance with the principles of the Declaration of Helsinki. The study was approved by the Institutional Review Board Ethics Committee (2020-440-1).

### 2.2. Design

A mixed concurrent triangulation design (dItrIaC) was carried out according to Creswell et al., Hernández et al., and García et al. [20,21,22]. These authors indicated that this design aims to confirm results, cross-validate results between quantitative and qualitative data, and apply the advantages of each method. Both quantitative and qualitative phases have equal importance in the research. In addition, both methods were applied at the same time, thus data collection and analysis were performed at the same time. Regarding the quantitative phase, a descriptive and cross-sectional design was carried out. Regarding the qualitative phase, a phenomenological design was used, in which the immediate subjective experience was analyzed as the basis of knowledge, whilst phenomena were studied from the perspective of the participants, and their referential framework was preserved. In addition, in the qualitative phase, interest was maintained in knowing how people experience and interpret the social world, which they construct in interaction through language [23].

### 2.3. Data Collection

All participants were included in the quantitative and in the qualitative analysis. Sociodemographic data (age, sex, time since HF diagnosis, education, marital status) were obtained by clinical interview through a trained researcher.

In the quantitative phase, the following outcomes were evaluated before and during COVID-19 confinement:(1)Well-being was assessed using the Cantril Ladder of Life [19], a single-item indicator with a ladder of steps numbered from 0 to 10 (0 = the worst possible life, 10 = the best possible life). Participants answered on which step they stand at present. Cantril Ladder of Life validity and test-retest coefficients of 0.70 have been reported in previous studies in patients with acute coronary events. Several studies have previously used this scale in HF patients [24,25,26,27,28].(2)Physical activity was evaluated using the International Physical Activity Questionnaire (IPAQ). It contains seven items for identifying frequency and duration of low, moderate, and vigorous PA as well as inactivity during the past week. Frequency is measured in days and duration in hours and minutes. The answers to the questions were transformed into metabolic equivalent of task (MET-minutes). The total PA score is the sum of vigorous, moderate, and walking PA scores. Typical IPAQ correlations with an accelerometer were 0.80 for reliability [29]. Several studies have previously used this questionnaire in HF patients [30,31,32].

Regarding the qualitative phase, semi-structured interviews were performed. Participants were invited to share information before and during COVID-19 confinement, as well as examples of situations they experienced. Table 1 shows the questions of the semi-structured interview performed.

### 2.4. Sample Size Calculation

Anticipating medium-sized differences in well-being and PA assessed before and during COVID-19 confinement in older adults with HF, the a priori power analysis (within-between interaction ANOVA) with two independent groups (octogenarians vs. non-octogenarians) and two measurement times (before vs. during COVID-19 confinement) yielded a needed total sample size of 98 participants using the following settings: f = 0.25, alpha = 0.05 (*p*-value), 1 − beta = 0.80 (power), correlation among repeated measures = 0.50.

### 2.5. Data Analysis

The statistical analysis in the quantitative phase was performed using SPSS version 26.0 (IBM SPSS, Inc., Chicago, IL., USA). An ANOVA test was used to explore differences between time measurements (i.e., before and during COVID-19 confinement) in the well-being and PA variables in the entire sample. Additionally, a two-factor mixed multivariate analysis of variance (MANOVA) with a between-subjects factor “groups by age” with two categories (i.e., non-octogenarians (<80 years) and octogenarians (≥80 years)) and a within-subject factor “time measurements” with two categories (i.e., before and during COVID-19 confinement) was performed in the abovementioned variables. Post-hoc analyses were requested using the Bonferroni correction. We evaluated the assumption of homoscedasticity using Levene’s test and the sphericity using Mauchly’s test. Partial eta-squared (η^2^p) values within the repeated measures ANOVA were used to express the effect size (i.e., small ≥ 0.01, medium ≥ 0.06, large ≥ 0.14). The α level was set equal to or less than 0.05 for all tests.

The data analysis in the qualitative phase was supported by computer-assisted qualitative data analysis software (CAQDAS) [33], specifically with the use of NVivo software version 12.0 (QSR International, Inc., Burlington, MA, USA). The treatment of the data followed the classical qualitative data analysis system [34,35]. This model involved the following steps [36]:(1)Data reduction. Information was divided into grammatical content units (paragraphs and sentences). Inductive content analysis (elaborating categories from the reading and analysis of the collected material without taking into consideration the initial categories) and deductive content analysis (categories are established a priori whilst the researcher adapts each unit to an already existing category) were performed. The assessment of content belonging to the corresponding category/subcategory was performed based on two levels, intracoder and intercoder, until agreement was reached among the members of the research team [37].(2)Layout and grouping. Different graphic resources and information were obtained using CAQDAS as follows: relationships and deep structure of the text [38], graphic representations or visual images of the relationships between concepts [39], and matrices/double-entry tables in which verbal information was included according to the aspects specified by rows and columns [34]. For the calculation of the analysis of the frequency of concurrence of the categories and subcategories, the NVivo coding matrix tool was been used. A matrix was made for each category, taking into account that the subcategories were placed in the rows, whilst the classification of octogenarian/non-octogenarian was placed in the columns. The percentages of each row were calculated based on the total cell references of each subcategory.(3)Obtention of results and verification of conclusions. This phase involved the use of metaphors and analogies, as well as the inclusion of vignettes and narrative fragments, culminating with the aforementioned triangulation strategies. For textual data, description, interpretation, code counting, concurrence, comparison, and contextualization were performed. For data transformed into numerical values, statistical techniques, comparison, and contextualization were performed.

## 3. Results

A total of 120 HF patients were assessed before and during COVID-19 confinement, of whom 60.80% were male and 39.20% were women. Mean (SD) age was 74.16 ± 12.90 years old. The sample consisted of non-octogenarians (<80 years old) (55.83% of the sample) and octogenarians (≥80 years) (44.16 % of the sample). All subjects had a diagnosis of HF, with a mean (SD) time of evolution of 78.73 ± 94.21 months. Table 2 shows the sociodemographic characteristics of the sample.

Table 3 shows the results of well-being and physical activity before and after confinement. The sample reported significantly lower well-being (*p* < 0.001) during confinement than before confinement. Similarly, lower levels of light PA (*p* < 0.001), moderate PA (*p* < 0.001), and total PA (*p* < 0.001) were observed during confinement when compared to before confinement. In addition, sedentary time was higher during confinement than before confinement (*p* < 0.001). However, no significant differences were found in vigorous PA (*p* > 0.05).

Two-factor repeated measures MANOVA revealed significant interaction effects on well-being and PA variables (F(5,114) = 3.33, *p* = 0.01, η^2^p = 0.89). Five univariate variables showed non-significant interaction effects for “group by age” and “time measurements”, including light PA, moderate PA, vigorous PA, sedentary time, and total PA (*p* > 0.05). However, the well-being univariate variable was found to have significant interaction effects for only “group by age” and “time measurements” (F(1,118 = 14.56, *p* < 0.001, η^2^p = 0.97). Table 4 shows the comparisons of well-being and physical activity in octogenarians vs. non-octogenarians before and after confinement. Post-hoc analysis showed that before confinement, the levels of light, moderate, vigorous, and total PA, as well as sedentary time were similar between both groups (*p* > 0.05). However, during confinement, octogenarians showed significantly lower values of well-being (*p* = 0.02) and moderate PA (*p* = 0.04), as well as higher sedentary time (*p* = 0.03) than non-octogenarians. Regarding between-time measurements comparisons, there were significant differences before and during confinement in well-being (*p* < 0.001, respectively), light PA (*p* < 0.001, respectively), moderate PA (*p* = 0.001 and *p* = 0.02, respectively), sedentary time (*p* < 0.001, respectively), and total PA (*p* < 0.001, respectively) in both non-octogenarians and octogenarians. However, there were no significant differences by time measurements in vigorous PA (*p* > 0.05) in any of the two groups.

Regarding the qualitative data analysis, five categories were obtained: (i) alterations in well-being; (ii) changes in social support; (iii) attention to HF symptoms; (iv) assistance needs; and (v) suggestions to improve care. The identified categories were divided into 32 subcategories. The identified categories and subcategories are described as follows:(i)Alterations in well-being. This category defines the main areas of well-being affected by COVID-19. It is composed of the following six subcategories: (1) cognitive; (2) the confinement has been positive on his/her well-being; (3) the confinement has not changed his/her routines; (4) emotional; (5) physical; (6) social.(ii)Changes in social support. This category shows the main changes in social support for people with HF during COVID-19. It is composed of six subcategories: (1) receives care from institutions; (2) receives care from his/her relatives; (3) difficulty in communication; (4) no family visits; (5) receives outside assistance; (6) reduced social contact.(iii)Attention to HF symptoms. This category highlights the main characteristics related to the attention to HF symptoms during confinement. It is composed of ten subcategories: (1) self-diagnosis of his/her health condition; (2) avoids social contact; (3) maintains healthy habits; (4) increased dependence; (5) fear of COVID-19; (6) no changes due to COVID-19; (7) does not follow doctor’s treatment; (8) concern for the health of his/her relatives; (9) concern for his/her own health; (10) health problems.(iv)Assistance needs. This category defines the needs of care and highlights the main needs or care requirements of HF patients during confinement. This category is composed of two subcategories and four contexts that define assistance needs: (1) no assistance needs; (2) if had assistance needs: (2a) assistance from family members; (2b) outside assistance (ambulance, telephone assistance, telecare button, caregiver, person for household chores and shopping, cardiac rehabilitation, neighbors); (2c) needs more assistance than received; (2d) total dependency.(v)Suggestions to improve care. This category highlights the suggestions of HF patients in order to improve their care during COVID-19 confinement. It is composed of the following eight subcategories: (1) help from politicians and from institutions; (2) demonstrate care to the patient; (3) increase the availability of physicians; (4) increase efficiency and patient care; (5) no suggestions; (6) speed in caring for patients; (7) receive written feedback from telephone consultations; (8) satisfied with the assistance received.

In the qualitative results, the most reported subcategories in each analyzed category are described. Findings are shown by verbatim excerpts from the interviews.

Regarding well-being, the 52.50% of the sample considered that COVID-19 altered their well-being to a great extent. Figure 1 shows differences between octogenarians vs. non-octogenarians in alterations in well-being caused by COVID-19 confinement. Alterations in emotional factors stand out in both groups, according to participant 45: “because it is very scary that people dye because of COVID-19. Our life has changed because we can’t go calmed down the street, nor when we meet family or friends” and participant 5: “I started teleworking and it has been overwhelming because we had more work than ever”. Moreover, physical alterations were highlighted, as participant 3 stated: “regarding physical level, I get more tired when making any effort”. It should be noted that non-octogenarians reported that confinement had less impact on their habits, as indicated by participant 164: “the confinement seemed very good to me because I don´t usually go out and I am with my husband and my daughter, so it is fine for me” and participant 133: “regarding confinement, I could climb stairs as I did before”.

On the other hand, 30% of the individuals reported that social support decreased during confinement. Figure 2 shows differences between octogenarians vs. non-octogenarians in changes in social support during COVID-19 confinement. In this regard, the most important change in octogenarians was the increase of care provided by their relatives, as shown by participant 1: “my family members are much more concerned about me and about the fact that I comply with the treatments” and participant 63: “I am living with my daughter, because due to confinement, I needed her help and care and I did not want to be alone”. In contrast, non-octogenarians perceived they did not received care during confinement from physicians, or they did not receive enough information in this regard; according to participant 187: “No doctor has come to my house to give information to me regarding suggestions of care and aspects to take into account related to confinement“ and participant 156: “I have not received any support”.

In addition, 50% of the individuals reported that their attention to HF symptoms increased during confinement. Figure 3 shows differences between octogenarians vs. non-octogenarians in attention to HF symptoms during COVID-19 confinement. There was a greater concern for health in non-octogenarians, as shown by participant 101: “I worry more about my health and I don’t want to get worse” and participant 196: “a lot of concern for me and for my wife. We don’t want to get sick, so we take more care of ourselves and pay more attention to our symptoms”. Similarly, there was a greater fear of COVID-19 in non-octogenarians, as indicated by participant 28: “I am scared, I am very afraid of COVID-19 [...] I am a patient at risk” and participant 87: “I pay more attention to my symptoms because of fear and because I have more time”. Although only slightly, octogenarians showed a greater concern for avoiding social contact, as expressed by participant 98: “I avoid contact with friends, even with relatives” and participant 178: “I only go out to the street one day a week and by car”.

Regarding assistance needs of HF patients, 60% of individuals in the sample were not satisfied with the assistance received. As shown in Figure 4, both groups needed assistance, with a greater predominance in octogenarians, as reported by participant 1: “I have a 24-h caregiver, she runs all my errands, goes shopping, helps me in the bath, checks my pills” and participant 23: “another caregiver to clean the house and go shopping”. On the other hand, non-octogenarians considered that they needed more help than they received, as reported by participant 19: “I need more attention because my legs that are swelling more, and I don’t understand why” and participant 229: “the doctor is more available. The general practitioner does not take care of me, he only gives medication to me, we cannot go to the outpatient clinic”.

Regarding suggestions to improve care (Figure 5), half of the individuals had no suggestions in this regard. Non-octogenarians suggested the need of offering a greater efficiency and attention to HF patients, as participant 39 assured: “I need to continue with the rehabilitation sessions that were cancelled due to the confinement. The suggestion I make is that those of us who are more dependent and who need more attention should get it. They leave us alone at home as if nothing happened, and that is bad for us. I suggest more attention, that the nurse or physiotherapist comes to check that I am doing well. Self-care starts with good information and making sure that I am doing well, then I will be able to do it alone” and participant 114: “they should call us to follow up our situation, it is hard for me to follow the recommendations”. However, those aged <80 suggested a lower availability of doctors, as suggested by participant 189: “now that I have pain and doctors do not attend me” and participant 98: “maybe if doctors were more available they could solve my doubts”, as well as reporting satisfaction with the assistance they were offered, as indicated by participant 23: “I am happy with the assistance received, and I send encouragement to the health workers” and participant 101: “very competent people, I am very happy with the treatment”.

Finally, the key points of the qualitative data analysis were that most of the individuals sampled considered that well-being decreased during confinement, 30% reported that social support decreased, 50% increased their attention to symptoms, and 60% were not satisfied with the assistance received. Octogenarians were more affected during lockdown than non-octogenarians in terms of well-being, attention to symptoms, and support needs.

## 4. Discussion

The findings of this study showed that well-being and PA levels of older adults with HF decreased during COVID-19 confinement. Regarding the comparison between octogenarians and non-octogenarians, octogenarians reported lower well-being, higher sedentary time, and lower levels of moderate PA than non-octogenarians during confinement. Moreover, octogenarians were more severely impacted during confinement than non-octogenarians in terms of exhibiting lower well-being. These results are in accordance with the patient-reported experiences, where octogenarians and non-octogenarians explained that confinement affected their well-being, especially regarding emotional status and physical fitness. Furthermore, octogenarian patients of this study reported that confinement affected several aspects of their daily lives. These results are especially important because previous studies have found that patients with HF are at a higher risk of developing mental health problems and seem to have a lower capacity to enjoy daily activities compared with people without HF symptoms [13]. In addition, in the study of Rantanen et al. [40], people more than 85 years old, both men and women, reported significantly lower quality of life during COVID-19 confinement. In addition, those with lower cognitive functioning, lower emotional stability, and living alone may be at risk of poorer self-reported mental and physical health [41]. Therefore, it is important to stress that nurses and other cardiac providers should identify vulnerabilities in sustained HF self-care behaviors and well-being among older adults, especially among octogenarians, to improve their well-being and physical activity and to improve satisfaction with the care they receive [19].

Furthermore, the decrease in PA and the increase in sedentary time that we found in older adults with HF during confinement, are in line with the results reported by Vetrovsky et al. [11], Angelo-Brasca et al. [42], and Caraballo et al. [12], who also found a reduction of PA levels in HF patients. It is important to highlight the decrease of PA levels since although physical condition was not evaluated, previous studies suggest that the decrease of PA levels could lead to an important deterioration of physical fitness, which is an important predictor of HF morbidity and mortality in this population [43,44]. In turn, a higher functional status is a predictor of an enhanced quality of life in octogenarians [45]. Therefore, the promotion of active aging seems to be a key point in order to improve or sustain quality of life, especially during social distancing [40].

A lower attention to their HF symptoms and higher assistance needs were found in octogenarians. However, non-octogenarians reported lower social support than octogenarians. In our sample, it seems that octogenarians reported an increase of care provided by their relatives, which may have influenced their social support; thus, octogenarians presented higher social support than non-octogenarians. In this regard, our results are not in line with those reported by Golden et al. [46], who found that loneliness increased with age. In addition, fear of contagion implied that patients preferred telephone visits, although both groups, especially octogenarians, reported that they needed assistance. These results are similar to those obtained by Mcilvennan et al. [3], who reported patients’ expressions of fear, reluctance to visit the hospital, and lower early symptom reporting. However, our results evidenced that participants were mostly satisfied with the care they received. In contrast, Raman and Vyselaar [10] reported that patients preferred in-person visits since telehealth visits were considered as presenting inferior quality due to the lack of physical examination, emotional detachment from care providers, and general unfamiliarity. Independently of participants´ opinions, telehealth models for outpatients with HF have been demonstrated to be safe and seem not to increase mortality, suggesting that telehealth outpatient visits in patients with HF could be safely incorporated into clinical practice [14,17].

Given the lack of studies that use a mixed-methods methodology to investigate well-being, PA, social support, attention to symptoms, assistance needs, and suggestions to improve care in older patients with HF during COVID-19 confinement, this study aimed to fill this gap for a better understanding of this condition. The comparison by age in octogenarians vs. non-octogenarians and the qualitative approach, which involved in-depth interviews with HF patients, led to the collection of relevant data not normally discussed or shared in healthcare research.

However, there are several limitations in this study. First, the study was performed in a single center; therefore, results may not be generalizable to the entire older population with HF. Additionally, the findings may not be representative of all patients, since our results can only be generalized to countries with similar restrictions to those established in Spain, such as Italy, France, or the Czech Republic. Second, the small sample size, although not unusual in qualitative research that requires extensive and detailed analysis of each patient, may not be representative of people with HF in Spain. Nevertheless, further studies should be performed with larger sample sizes. Future studies should also investigate the consequences of this reduction of PA and implement appropriate protocols to ensure good health outcomes in older adults, considering well-being, social support, attention to HF symptoms, and assistance needs during critical periods.

## 5. Conclusions

Well-being and PA levels decreased during COVID-19 confinement in older adults with HF. Moreover, octogenarians were more severely impacted during confinement than non-octogenarians in terms of lower well-being, lower attention to their HF symptoms, and higher assistance needs, although non-octogenarians reported lower social support than non-octogenarians. Thus, the development of remote monitoring strategies is needed in older adults with HF in order to maintain an adequate level of PA and control health outcomes in critical periods, especially for octogenarians with HF.

## Figures and Tables

**Figure 1 ijerph-19-15316-f001:**
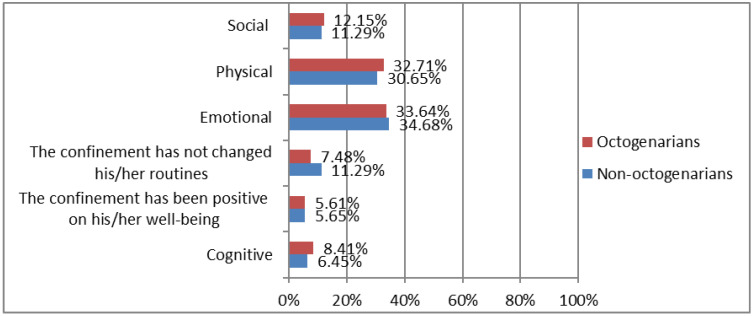
Differences between octogenarians vs. non-octogenarians in alterations in well-being caused by COVID-19 confinement. Octogenarians (≥80 years; *n* = 53). Non-octogenarians (<80 years; *n* = 67).

**Figure 2 ijerph-19-15316-f002:**
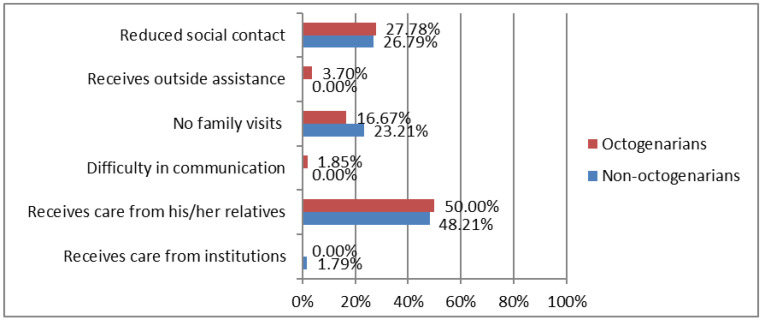
Differences between octogenarians vs. non-octogenarians in changes in social support during COVID-19 confinement. Octogenarians (≥80 years; *n* = 53). Non-octogenarians (<80 years; *n* = 67).

**Figure 3 ijerph-19-15316-f003:**
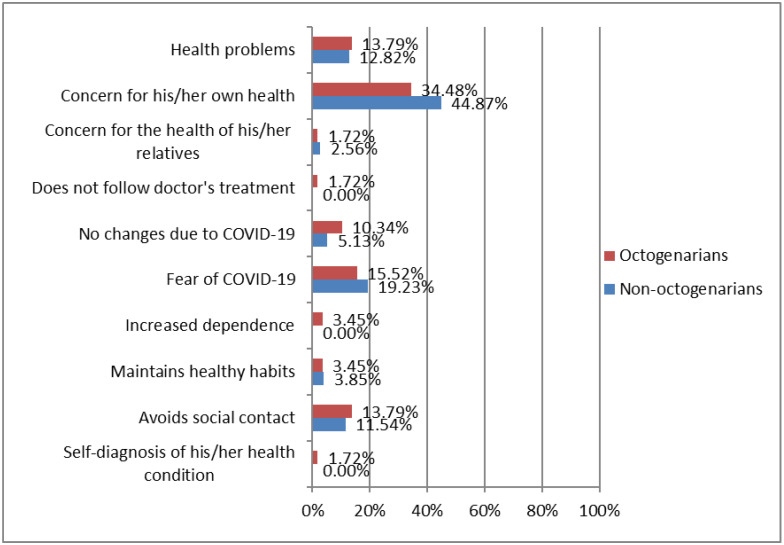
Differences between octogenarians vs. non-octogenarians in attention to heart failure symptoms during COVID-19 confinement. Octogenarians (≥80 years; *n* = 53). Non-octogenarians (<80 years; *n* = 67).

**Figure 4 ijerph-19-15316-f004:**
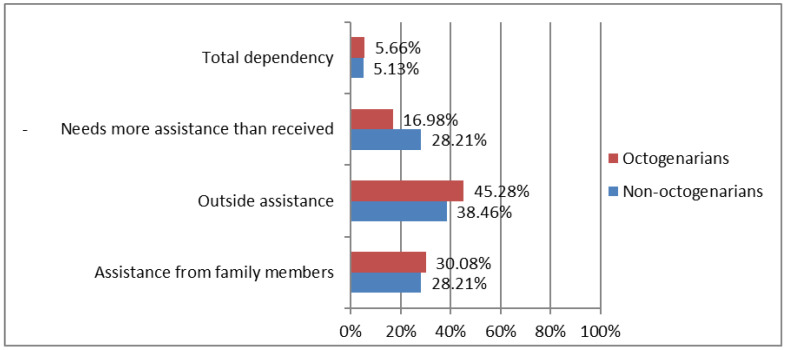
Differences between octogenarians vs. non-octogenarians in in assistance needs during COVID-19 confinement. Octogenarians (≥80 years; *n* = 53). Non-octogenarians (<80 years; *n* = 67).

**Figure 5 ijerph-19-15316-f005:**
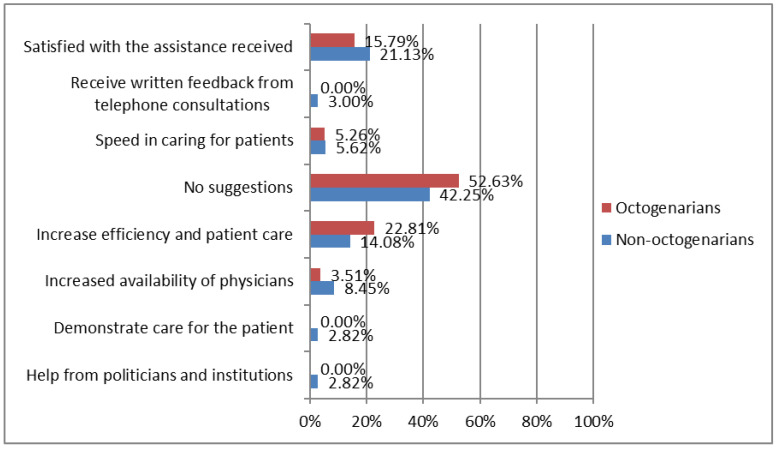
Differences between octogenarians vs. non-octogenarians in suggestions to improve care during COVID-19 confinement. Octogenarians (≥80 years; *n* = 53). Non-octogenarians (<80 years; *n* = 67).

**Table 1 ijerph-19-15316-t001:** Questions of the semi-structured interview.

1. Has your well-being been altered during COVID-19 confinement? Please explain. 2. How has the social support (family, friends, neighbors) you receive changed during the COVID-19 confinement? Please explain. 3. How has your attention to heart failure symptoms changed due to the COVID-19 confinement? Please explain. 4. What assistance needs do you have now during the COVID-19 confinement? Please explain. 5. Do you have any suggestions or advice on how health and healthcare services can assist you in your self-managed care for your heart failure? Please explain.

**Table 2 ijerph-19-15316-t002:** Sociodemographic characteristics of the sample.

Variable	Mean ± SD/Frequency (Percentage)
Age (years)	74.16 ± 12.90
Non-octogenarians (<80 years)	67 (55.83%)
Octogenarians (≥80 years)	53 (44.16%)
Sex	
Male	73 (60.83%)
Female	47 (39.17%)
Time since diagnosis (months)	78.73 ± 94.21
Education	
Primary	79 (65.83%)
Secondary	17 (14.17%)
University	24 (20.00%)
Marital status	
Single	5 (4.17%)
Married	81 (67.50%)
Divorced	2 (1.67%)
Widow	32 (26.67%)

Data shown as mean ± standard deviation (SD) for quantitative variables and absolute frequency (percentage) for ordinal variables.

**Table 3 ijerph-19-15316-t003:** Well-being and physical activity of older patients with HF before and during COVID-19 confinement.

Variable	Before COVID-19 Confinement (*n* = 120)	During COVID-19 Confinement (*n* = 120)	*p*-Value
Well-being	7.5 ± 1.57	5.98 ± 2.14	*p* < 0.001 *
Physical activity (METS-minute)			
Light PA	764.71 ± 826.08	103.59 ± 376.77	*p* < 0.001 *
Moderate PA	6.19 ± 13.59	1.49 ± 3.39	*p* < 0.001 *
Vigorous PA	1.68 ± 14.81	0.00 ± 0.00	*p* = 0.26
Sedentary time	7.03 ± 2.41	9.39 ± 2.77	*p* < 0.001 *
Total PA	772.57 ± 828.96	105.08 ± 376.90	*p* < 0.001 *

Data shown as mean (M) and standard deviation (SD). *: *p* < 0.05 between time measurements. PA = physical activity.

**Table 4 ijerph-19-15316-t004:** Well-being and physical activity of octogenarians vs. non-octogenarians HF patients before and during COVID-19 confinement.

Variable	Non-Octogenarians (<80 Years) (*n* = 67)	Octogenarians (≥80 Years) (*n* = 53)	*p*-Value between Groups
Well-being			
Before COVID-19 confinement	7.46 ± 1.65	7.55 ± 1.47	*p* = 0.77
During COVID-19 confinement	6.37 ± 2.12	5.49 ± 2.09	*p* = 0.02 *
*p*-value between time measurements	*p* < 0.001 *	*p* < 0.001 *	
Physical activity (METS-minute)			
Light PA			
Before COVID-19 confinement	844.93 ± 967.35	663.29 ± 595.96	*p* = 0.23
During COVID-19 confinement	104.02 ± 285.64	103.05 ± 470.43	*p* = 0.99
*p*-value between time measurements	*p* < 0.001 *	*p* < 0.001 *	
Moderate PA			
Before COVID-19 confinement	7.09 ± 14.44	5.05 ± 12.47	*p* = 0.42
During COVID-19 confinement	2.05 ± 3.93	0.80 ± 2.40	*p* = 0.04 *
*p*-value between time measurements	*p* = 0.001 *	*p* = 0.02 *	
Vigorous PA			
Before COVID-19 confinement	2.75 ± 19.72	0.33 ± 2.20	*p* = 0.38
During COVID-19 confinement	0.00 ± 0.00	0.00 ± 0.00	*p* = 1.00
*p*-value between time measurements	*p* = 0.13	*p* = 0.87	
Sedentary time			
Before COVID-19 confinement	6.58 ± 2.20	7.58 ± 2.56	*p* = 0.06
During COVID-19 confinement	8.91 ± 2.261	10.00 ± 3.22	*p* = 0.03 *
*p*-value between time measurements	*p* < 0.001 *	*p* < 0.001 *	
Total PA			
Before COVID-19 confinement	854.77 ± 971.31	668.66 ± 596.16	*p* = 0.39
During COVID-19 confinement	106.06 ± 286.01	103.84 ± 470.37	*p* = 0.15
*p*-value between time measurements	*p* < 0.001 *	*p* < 0.001 *	

Data shown as mean (M) and standard deviation (SD). *: *p* < 0.05. PA = physical activity.

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
