# Peer review of "Well-Being, Physical Activity, and Social Support in Octogenarians with Heart Failure during COVID-19 Confinement: A Mixed-Methods Study"

_ijerph, 2022, doi:10.3390/ijerph192215316_

Round 1

Reviewer 1 Report

This study is important because heart failure is a major clinical and public health problem

1.      Although the authors claim about few qualitative or mixed-methods studies have been conducted comparing this study, they did not show any evidence.

2.      Although the authors claim that the sample consisted of 120 participants but there was no description of the appropriate sample size formula.

3.      The development of questionnaires (previous studies, design, pretest, etc.) was not also described clearly.

4.      How were sociodemographic data (age, sex, time since HF diagnosis, education, marital status) selected by variable selection using the appropriate technique is important for any statistical analysis.

5.      Authors are suggested to use an appropriate advance multivariate statistical model for analysis.

6.      Ethics approval should be described in detail.

7.      The topic is too large to read.

Author Response

First, authors would like to thank Reviewer 1 for the time spent in suggesting changes and improvements to our article. According to their suggestions, we believe the paper has been improved and clarified: the following responses have been prepared to address the reviewers’ comments in a point-by-point fashion.

COMMENT #1: Although the authors claim about few qualitative or mixed-methods studies have been conducted comparing this study, they did not show any evidence.

AUTHORS RESPONSE #1: We thank the reviewer for the comment. We have added the references and an explanation of the referred studies in this regard.

COMMENT #2: Although the authors claim that the sample consisted of 120 participants but there was no description of the appropriate sample size formula.

AUTHORS RESPONSE #2: According with the reviewer suggestion, we have added the sample size calculation in the manuscript. “For computing sample size, we considered that our study was composed of two groups and two measurements times, and we set a power of 80% and an effect size of f2=0.25. This generated a minimum sample size of 120 participants.”

COMMENT #3: The development of questionnaires (previous studies, design, pretest, etc.) was not also described clearly.

AUTHORS RESPONSE #3: The authors thank the reviewer the suggestions and we have improved the explanation of these matters in the material and methods subsection, as well as added several references supporting the use of this questionnaires in HF patients.

COMMENT #4: How were sociodemographic data (age, sex, time since HF diagnosis, education, marital status) selected by variable selection using the appropriate technique is important for any statistical analysis.

AUTHORS RESPONSE #4: We thank the reviewer for the comment. These outcomes are usually measured as sociodemographic outcomes in studies with HF patients (Klompstra L, Deka P, Almenar L, Pathak D, Muñoz-Gómez E, López-Vilella R, Marques-Sule E. Physical activity enjoyment, exercise motivation, and physical activity in patients with heart failure: A mediation analysis. Clin Rehabil. 2022 Oct;36(10):1324-1331; Deka P, Almenar L, Pathak D, Klompstra L, López-Vilella R, Marques-Sule E. Depression mediates physical activity readiness and physical activity in patients with heart failure. ESC Heart Fail. 2021 Dec;8(6):5259-5265.). Sociodemographic outcomes were obtained by clinical interview through a trained researcher.

COMMENT #5: Authors are suggested to use an appropriate advance multivariate statistical model for analysis.

AUTHORS RESPONSE #5: We thank the reviewer for his comment. We have performed the inferential analyses with the Student-t test and ANOVA to analyze between-group and between-time differences as appropriate. The changes can be seen in the methods section "Data analysis" and in the results section (in the text of the manuscript, and also in Tables 3 and 4).

COMMENT #6: Ethics approval should be described in detail.

AUTHORS RESPONSE #6: We thank the reviewer for the comment. All the information about the ethics approval process, including number of registration and data of approval, has been described with more detail in the manuscript, and has also been included in the section “Institutional Review Board Statement” at the end of the manuscript.

COMMENT #7: The topic is too large to read.

AUTHORS RESPONSE #7: We agree that the title was too long previously, then we have changed the title of the manuscript in order to clarify and shorten it.

Reviewer 2 Report

-Abtract: should check the objectives of this study, it not related to the objectives in content(3 objectives)

-Introduction : Why study compare between octogenarians and non-octogenarians , should mention and give the reasons in this part

- The  objectives: it present 3 thing ,not related to abtract part

-Material and methods: clearefy about number of participants in qualitative part, and inclusion criterea show ages of paticipants 60 years and above ,but data collection part researcher show well - being tool, it used for more than 65 yeara and above,why!

-Results part : Table 2 should explain the main results before present the table

-the results of qualitative data should  start that the qualitative data found that... and should be summary of key point at the end.

- Figure no.1-5 should present no.of participants, presentation only % not clear 

- Unclear about qualitative data ex. no. 63 in line 200,187 in line 203, what is mean!

-Discussion part:  unclear about line261 ,why non-octogenarirn report lower social support than non-octogenariarn!

-Discussion part: should review, and follow by the 3 objectives

-Conclusion part: shuould be review check typing error

-Appendix A should delete, and some importent data sholdbe present in the results

- Supplementary materials should delete

Author Response

First, authors would like to thank Reviewer 2 for the time spent in suggesting changes and improvements to our article. According to their suggestions, we believe the paper has been improved and clarified: the following responses have been prepared to address the reviewers’ comments in a point-by-point fashion.

COMMENT #1: Abstract: should check the objectives of this study, it not related to the objectives in content (3 objectives)

AUTHORS RESPONSE #1: Thank you for the comment. We agree with reviewer 2 and we have changed the objective of the abstract.

COMMENT #2: Introduction: Why study compare between octogenarians and non-octogenarians, should mention and give the reasons in this part

AUTHORS RESPONSE #2: Thank you for the suggestion. We have completed this information in the Introduction section adding a justification, in order to give the reasons and importance of the comparisons between octogenarians and non-octogenarians.

COMMENT #3: The objectives: it present 3 thing, not related to abstract part.

AUTHORS RESPONSE #3: We thank the reviewer for this comment, we have changed the objective in the abstract.

COMMENT #4: Material and methods: clearefy about number of participants in qualitative part, and inclusion criterea show ages of paticipants 60 years and above ,but data collection part researcher show well-being tool, it used for more than 65 yeara and above,why!

AUTHORS RESPONSE #4: We thank the reviewer for the comment. The wording of the explanation of the Cantril Ladder of Life questionnaire has been improved to explain that in the present study the questionnaire was administered to all participants, and several references have also been added to the two quantitative questionnaires: Cantril Ladder of Life, and International Physical Activity Questionnaire. The number of participants was n=120, and all of them participated in the qualitative and in the quantitative analysis of this study. For this reason, no differentiation has been made in the number of participants.

COMMENT #5: Results part: Table 2 should explain the main results before present the table.

AUTHORS RESPONSE #5: According to the reviewer suggestion, we have added a short description of the sociodemographic characteristics of the sample before presenting the table.

COMMENT #6: The results of qualitative data should start that the qualitative data found that... and should be summary of key point at the end.

AUTHORS RESPONSE #6: The authors are very pleased to these suggestions and an explanatory paragraph has been added at the beginning of the qualitative results, including all the categories and subcategories obtained. Also, the key points of the qualitative data analysis have been added at the end of the results.

COMMENT #7: Figure no.1-5 should present no.of participants, presentation only % not clear.

AUTHORS RESPONSE #7: Thank you for the comment. However, we have clarified in the methods section that for the calculation of the analysis of the frequency of concurrence of the categories and subcategories, the NVivo coding matrix tool has been used. A matrix was made for each category, taking into account that the subcategories were placed in the rows, whilst the classification octogenarian/non-octogenarian was placed in the columns. The percentages of each row are calculated based in the total de references of each subcategory. However, the number of participants in each group (octogenarians, non-octogenarians) has been included in figures 1 to 5: Octogenarians (≥ 80 years; n = 53). Non-octogenarians (< 80 years; n = 67). 

COMMENT #8: Unclear about qualitative data ex. no. 63 in line 200,187 in line 203, what is mean!

AUHORS RESPONSE #8: We thank the reviewer for the comment. In order to clarify these results referred, we have added more information to quote from participant number 63 and from participant number 187.

COMMENT #9: Discussion part:  unclear about line 261 ,why non-octogenarirn report lower social support than non-octogenariarn!

AUHORS RESPONSE #9: Thanks for the appreciation. In our sample we obtained this finding, and we believe it may be due to the fact that octogenarians reported an increase of care provided by their relatives, which may have influenced in their social support, thus octogenarians presented higher social support than non-octogenarians. We have included information in the discussion in this regard.

COMMENT #10: Discussion part: should review, and follow by the 3 objectives

AUHORS RESPONSE #10: We thank the reviewer for the comment. The discussion part has been reorganized coherently in the order of the 3 main objectives.

COMMENT #12: Conclusion part: shuould be review check typing error

AUHORS RESPONSE #12: Thank you. The typo error has been corrected in the manuscript.

COMMENT #13: Appendix A should delete, and some importent data sholdbe present in the results

AUHORS RESPONSE #13:  Thanks for the appreciation. The Appendix A has been deleted as suggested by the reviewer, and the different categories and subcategories have been included in the body of the article.

COMMENT #14: Supplementary materials should delete

AUHORS RESPONSE #14:  We thank the reviewer for the comment and have deleted the Appendix.

Reviewer 3 Report

Thank you for the opportunity to review this manuscript. This study focuses on older adults with heart failure to analyze how COVID-19 confinement affected them. The primary aim of this study was to compare well-being and physical activity before and during COVID-19 confinement in older adults with heart failure. I like this manuscript and believe that it provides possible solutions to maintain an adequate level of physical activity and control health outcomes in critical periods.

In general, the manuscript is well-written, and the topic is relevant. However, there are a few questions about the study that need to be addressed and added to the paper:

-page 2 line 79 Please explain how do you get the sample size for this study.

-Page 2 line 80 “...were assessed between November 2020 and April 2020.” Please check the year.

-In table 2, the total numbers in the variable Marital status is 119. Please double-check the numbers.

-It is not clear how to define before COVID-19 confinement and during COVID-19 confinement in the manuscript. Please add the information in the Materials and Methods section.

-Materials and Methods- Data analysis. I would suggest adding information on the reason why choose the Mann-Whitney U tests and the Wilcoxon test. Please clarify which method was used in Tables 3 and 4, Mann-Whitney U tests or Wilcoxon test.

-Please check if this study involves multiple comparisons.

-In Tables 3 and 4, did each participant assess by Cantril Ladder of Life (Well-being) and International Physical Activity Questionnaire (Physical activity) twice? One is before COVID-19 and the other one is during COVID-19. Or out of 120 participants, some participants completed the well-being and physical activity assessment before COVID-19, and the rest of the participants completed the well-being and physical activity assessment during COVID-19. Please clarify.

-There is no explanation in the Materials and Methods section about how you calculate the percentages of figure 1 - figure 5. How do you get the denominator and numerator? I would recommend adding the above details in the Materials and Methods section.

-In figure 1, the social, physical, emotional, and cognitive are neutral while in the supplementary material these subcategories are negative. Please be consistent with the polarity of subcategories. Similar to other figures.

-Page 6 line 196 “On the other hand, one third of the sample reported that social support decreased …” and line 208 “More than a half of the sample considered that attention to HF symptoms increased …” I couldn’t find the information of “one third” and “More than a half” in figure 2 and figure 3. If authors plan to calculate the percentages at the category level, I would suggest adding the details of the denominator and numerator in the Materials and Methods section and the numbers in the results section.

-Page 10 line 330 “... especially for octogenarians with HF.6. Patents” Please check “6. Patents”.

Author Response

First, authors would like to thank Reviewer 3 for the time spent in suggesting changes and improvements to our article and we sincerely appreciate the positive feedback received. According to their suggestions, we believe the paper has been improved and clarified: the following responses have been prepared to address the reviewers’ comments in a point-by-point fashion.

COMMENT #1: Page 2 line 79 Please explain how do you get the sample size for this study.

AUTHORS RESPONSE #1: According with the reviewer suggestion, we have added the sample size calculation in the manuscript before the Data analysis section. “For computing sample size, we considered that our study was composed of two groups and two measurements times, and we set a power of 80% and an effect size of f2=0.25. This generated a minimum sample size of 120 participants.”

COMMENT #2: Page 2 line 80 “...were assessed between November 2020 and April 2020.” Please check the year.

AUTHORS RESPONSE #2: Thanks for the appreciation and we are sorry for the mistake. We have changed the year to “November 2019”.

COMMENT #3: In table 2, the total numbers in the variable Marital status is 119. Please double-check the numbers.

AUTHORS RESPONSE #3: We thank the reviewer for the note and we are sorry for the mistake. We have added the missing value in the marital status section of table 2.

COMMENT #4: It is not clear how to define before COVID-19 confinement and during COVID-19 confinement in the manuscript. Please add the information in the Materials and Methods section.

AUTHORS RESPONSE #4: We thank the reviewer for the comment. We agree with the reviewer that due to a typo error in the sentence “...were assessed between November 2020 and April 2020” it was impossible to understand this explanation. We hope that with the change of that sentence it has been clarified.

COMMENT #5: Materials and Methods- Data analysis. I would suggest adding information on the reason why choose the Mann-Whitney U tests and the Wilcoxon test. Please clarify which method was used in Tables 3 and 4, Mann-Whitney U tests or Wilcoxon test.

AUTHORS RESPONSE #5: We have clarified in the methods section that the inferential tests used were Student-t test and ANOVA. In addition, in the results section and in Tables 3 and 4 we present the differences between the groups and between time measurements as appropriate.

COMMENT #6: Please check if this study involves multiple comparisons.

AUTHORS RESPONSE #6: We thank the reviewer for the suggestion. Indeed, multiple comparisons have been made. In the “Data analysis” section we have detailed the following: “A two-factor mixed multivariate analysis of variance (MANOVA) with a between-subjects factor “groups by age” with two categories (i.e. non-octogenarians (< 80 years) and octogenarians (≥ 80 years)) and a withing-subject factor “time measurements” with two categories (i.e. before and during COVID-19 confinement) was performed in well-being and PA variables. Post-hoc analyses were requested using the Bonferroni correction.”

COMMENT #7: In Tables 3 and 4, did each participant assess by Cantril Ladder of Life (Well-being) and International Physical Activity Questionnaire (Physical activity) twice? One is before COVID-19 and the other one is during COVID-19. Or out of 120 participants, some participants completed the well-being and physical activity assessment before COVID-19, and the rest of the participants completed the well-being and physical activity assessment during COVID-19. Please clarify.

AUTHORS RESPONSE #7: In accordance with the reviewer’s request, we have clarified in both the methods section and the results section that all participants (n=120) were measured on two timepoints (i.e. before and during COVID-19 confinement).

COMMENT #8: There is no explanation in the Materials and Methods section about how you calculate the percentages of figure 1 - figure 5. How do you get the denominator and numerator? I would recommend adding the above details in the Materials and Methods section.

AUTHORS RESPONSE #8: We thank the reviewer for the suggestion. The procedure for calculating the frequencies of concurrence of each category is now shown in Materials and Methods: “For the calculation of the analysis of the frequency of concurrence of the categories and subcategories, the NVivo coding matrix tool has been used. A matrix was made for each category, taking into account that the subcategories were placed in the rows, whilst the classification octogenarian/non-octogenarian was placed in the columns. The percentages of each row are calculated based in the total de references of each subcategory.”

COMMENT #9: In figure 1, the social, physical, emotional, and cognitive are neutral while in the supplementary material these subcategories are negative. Please be consistent with the polarity of subcategories. Similar to other figures.

AUTHORS RESPONSE #9: We thank the reviewer for the suggestion. According to objective 3: explore well-being, social support, attention to HF symptoms, assistance needs, and suggestions to improve care during COVID-19 confinement in this population, we have explored the positive or negative aspects that HF patients report for each category, taking into account that the neutral values present a lack  of information for the qualitative analysis. Therefore, problems or difficulties of HF patients during COVID-19 confinement have been explored in the qualitative analysis.

COMMENT #10: Page 6 line 196 “On the other hand, one third of the sample reported that social support decreased …” and line 208 “More than a half of the sample considered that attention to HF symptoms increased …” I couldn’t find the information of “one third” and “More than a half” in figure 2 and figure 3. If authors plan to calculate the percentages at the category level, I would suggest adding the details of the denominator and numerator in the Materials and Methods section and the numbers in the results section.

AUTHORS RESPONSE #10: We thank the reviewer for the comment. The information has been included in the results section of the manuscript, in the qualitative results for each variable. In addition, as said before, we have explained in the methods section that we used the Nvivo program and how this process and the obtention of the percentages were performed.

COMMENT #11: Page 10 line 330 “... especially for octogenarians with HF.6. Patents” Please check “6. Patents”.

AUHORS RESPONSE #11: Thank you. The typo error has been corrected at the manuscript.

Round 2

Reviewer 1 Report

The authors should show the details sample size calculation and make sure that they calculated this sample size before the study was conducted. 

Author Response

We thank the reviewer for his/her suggestion. We have added the details of the sample size calculation process performed a priori with the G∗Power 3.1.7 for Windows software (l. 139-146). 

Reviewer 2 Report

Approved 

Author Response

Thank you very much for your approval for publication.

Reviewer 3 Report

The authors have satisfactorily addressed most of my concerns in my previous reviews. Below I have included suggestions that could further clarify the manuscript.

-line 30 “octogenarians=44.16%, non-octogenarians=55.83%).”. In table 2, the octogenarians=44.20%, non-octogenarians=55.80%. Please check and keep consistent.

-line 32 “reported lower well-being (p=0.02), higher sedentary time (p=0.04) and lower levels of moderate PA (p=0.04) during confinement”. Please check with line 207 and 208 to keep consistent.

-line 144 “An independent Student’s t-test was used to explore differences between time measurements (i.e. before and during COVID-19 confinement) in the well-being and PA variables in the whole sample”. The independent Student’s t-test is not correct. This is a repeated measure, where a single group of individuals is obtained and each individual is measured before and during COVID-19. They are not two independent groups. I would suggest reading the following paper and modify the method section:

The Differences and Similarities Between Two-Sample T-Test and Paired T-Test

https://www.ncbi.nlm.nih.gov/pmc/articles/PMC5579465/ 

-line 146 “a two-factor mixed multivariate analysis of variance (MANOVA) with a between-subjects factor “groups by age” with two categories (i.e. non-octogenarians (< 80 years) and octogenarians (≥ 80 years)) …” Did you try repeated measures ANOVA? Do you report the output of the MANOVA in the results section? I would suggest consult a statistician to use the appropriate method here.

- line 152 “The α level was set equal to or less than 0.05 for all tests” .It is not clear about the post-hoc tests α. This multiple-comparison post hoc correction is used when you are performing many independent or dependent statistical tests at the same time. For example, if you are running 20 simultaneous tests at α = 0.05, the correction would be 0.0025. In table 4, line 217 “p < 0.05 between groups and between time measurements”. The α for multiple-comparison could not be 0.05 here.  

-line 187 “a mean (SD) time of evolution of 78.73 ± 94.21 years”, I would suggest modifying the years to months.

-In table 2, please check the percentages. 

67/120=55.83%

53/120=44.17%

73/120=60.83%

-line 254 “most of the sample (52,50%) ...” I would suggest modifying “most” to another word. Please use 52.50% instead of 52,50%.

-line 284-286 “Octogenarians (≥ 80 years; n = 53). Non-octo-genarians (< 80 years; n = 67). In addition, 50% of the sample reported that their attention ...” The figure 2 title mixed with the new paragraph which start with “In addition, 50% of the sample reported that their attention”. Please check.

Author Response

Authors would like to thank reviewer for the time spent in suggesting changes and improvements to our article. According to their suggestions, we have prepared the comments in a point-by-point fashion.

COMMENT #1: line 30 “octogenarians=44.16%, non-octogenarians=55.83%).”. In table 2, the octogenarians=44.20%, non-octogenarians=55.80%. Please check and keep consistent. 

RESPONSE #1: Thanks for the suggestion. We have corrected the decimal numbers in table 2.

COMMENT #2: line 32 “reported lower well-being (p=0.02), higher sedentary time (p=0.04) and lower levels of moderate PA (p=0.04) during confinement”. Please check with line 207 and 208 to keep consistent.

RESPONSE #2: We are sorry for the mistake. We have corrected line 32 (p=0.03 instead of p=0.04).

COMMENT #3: line 144 “An independent Student’s t-test was used to explore differences between time measurements (i.e. before and during COVID-19 confinement) in the well-being and PA variables in the whole sample”. The independent Student’s t-test is not correct. This is a repeated measure, where a single group of individuals is obtained and each individual is measured before and during COVID-19. They are not two independent groups. I would suggest reading the following paper and modify the method section: The Differences and Similarities Between Two-Sample T-Test and Paired T-Test. https://www.ncbi.nlm.nih.gov/pmc/articles/PMC5579465/

RESPONSE #3: The reviewer is absolutely right. It is a repeated-measure design (doi: 10.4103/0975-9476.113872) in which subjects (a simple group of 120 participants) have been evaluated in two times (i.e. before and during COVID-19 confinement). We have corrected this in the data analysis section. In addition, after performing the analyses (ANOVA), it has been observed that the p-values remain as they were, except in the variable vigorous PA that goes from p = 0.22 to p = 0.26 (table 3).

COMMENT #4: line 146 “a two-factor mixed multivariate analysis of variance (MANOVA) with a between-subjects factor “groups by age” with two categories (i.e. non-octogenarians (< 80 years) and octogenarians (≥ 80 years)) …” Did you try repeated measures ANOVA? Do you report the output of the MANOVA in the results section? I would suggest consult a statistician to use the appropriate method here.

RESPONSE #4: A two-factor mixed multivariate analysis of variance (MANOVA) was applied to determine the interaction effect for the group by time measurements. The output of MANOVA with the F-test statistic, p-value and effect size expressed as partial eta-squared values (η2p) has been reported in the results section.

COMMENT #5: line 152 “The α level was set equal to or less than 0.05 for all tests” .It is not clear about the post-hoc tests α. This multiple-comparison post hoc correction is used when you are performing many independent or dependent statistical tests at the same time. For example, if you are running 20 simultaneous tests at α = 0.05, the correction would be 0.0025. In table 4, line 217 “p < 0.05 between groups and between time measurements”. The α for multiple-comparison could not be 0.05 here.  

RESPONSE #5: We thank the reviewer for his/her comment. We understand that performing multiple t-tests will lead the researcher to a higher probability of making a Type I error. However, according with Mary L. McHugh (doi: 10.11613/BM.2011.029), the great advantage of the multiple comparison analysis tests based on the Bonferroni method is that it reduces the probability of a Type I error by its limits on alpha inflation. In this case, it has been established that the mean difference is significant at the 0.05 level, as used in the literature (doi: 10.4103/0975-9476.113872). In table 4 we have specified the p-value of the comparisons between groups (i.e. non-octogenarians and octogenarians) in the last column, and the p-value for comparisons between time measurements in the last row of each variable.

COMMENT #6: line 187 “a mean (SD) time of evolution of 78.73 ± 94.21 years”, I would suggest modifying the years to months.

RESPONSE #6: We thank the reviewer the suggestion. We have changed years by months.

COMMENT #7: In table 2, please check the percentages. 67/120=55.83%. 53/120=44.17%. 73/120=60.83%.

RESPONSE #7: Following the reviewer’s suggestion, we have modified the percentages in table 2.

COMMENT #8: line 254 “most of the sample (52,50%) ...” I would suggest modifying “most” to another word. Please use 52.50% instead of 52,50%.

RESPONSE #8: We have made the changes proposed by the reviewer.

COMMENT #9: line 284-286 “Octogenarians (≥ 80 years; n = 53). Non-octogenarians (< 80 years; n = 67). In addition, 50% of the sample reported that their attention ...” The figure 2 title mixed with the new paragraph which start with “In addition, 50% of the sample reported that their attention”. Please check.

RESPONSE #9: Thanks for the correction.